# Hypernasality associated with basal ganglia dysfunction: evidence from Parkinson's disease and Huntington's disease

Michal Novotný[1], Jan Rusz[1,2], Roman Čmejla[1], Hana Růžičková[2], Jiří Klempíř[2,3] and Evžen Růžička[2]

[1] Department of Circuit Theory, Faculty of Electrical Engineering, Czech Technical University in Prague, Prague, CZ, Czech Republic
[2] Department of Neurology and Centre of Clinical Neuroscience, First Faculty of Medicine, Charles University in Prague, Prague, CZ, Czech Republic
[3] Institute of Anatomy, First Faculty of Medicine, Charles University in Prague, Prague, CZ, Czech Republic

## ABSTRACT

**Background**. Although increased nasality can originate from basal ganglia dysfunction, data regarding hypernasality in Parkinson's disease (PD) and Huntington's disease (HD) are very sparse. The aim of the current study was to analyze acoustic and perceptual correlates of velopharyngeal seal closure in 37 PD and 37 HD participants in comparison to 37 healthy control speakers.

**Methods**. Acoustical analysis was based on sustained phonation of the vowel /i/ and perceptual analysis was based on monologue. Perceptual analysis was performed by 10 raters using The Great Ormond Street Speech Assessment '98. Acoustic parameters related to changes in a 1/3-octave band centered on 1 kHz were proposed to reflect nasality level and behavior through utterance.

**Results**. Perceptual analysis showed the occurrence of mild to moderate hypernasality in 65% of PD, 89% of HD and 22% of control speakers. Based on acoustic analyses, 27% of PD, 54% of HD and 19% of control speakers showed an increased occurrence of hypernasality. In addition, 78% of HD patients demonstrated a high occurrence of intermittent hypernasality. Further results indicated relationships between the acoustic parameter representing fluctuation of nasality and perceptual assessment ($r = 0.51$, $p < 0.001$) as well as the Unified Huntington Disease Rating Scale chorea composite subscore ($r = 0.42$, $p = 0.01$).

**Conclusions**. In conclusion the acoustic assessment showed that abnormal nasality was not a common feature of PD, whereas patients with HD manifested intermittent hypernasality associated with chorea.

Corresponding author
Jan Rusz, rusz.mz@gmail.com

## INTRODUCTION

Considerable attention has been given to progressive neurodegenerative diseases affecting the basal ganglia such as Parkinson's disease (PD) and Huntington's disease (HD). Both PD and HD are terminal neurodegenerative diseases that elicit a variety of motor and non-motor manifestations, which significantly contribute to decreased quality of life (*Jankovic, 2008*; *Walker, 2007*). As PD and HD affect different regions of the basal ganglia, the manifestations differ between both diseases. In PD, damage of dopaminergic neurons in the substantia nigra and related dopamine depletion lead to debilitating loss of movement due to muscle rigidity, bradykinesia and resting tremor. In HD, damage to the striatum primarily results in extensive semi-directed, non-rhythmic movements termed chorea, dementia and psychiatric manifestations encompassing behavioral difficulties connected with lower emotional control and intense irritability (*Jankovic, 2008*; *Walker, 2007*).

The majority of both PD and HD patients manifest the motor speech disorder termed dysarthria (*Hartelius et al., 2003*; *Logemann et al., 1978*; *Rusz et al., 2014*), which is an impairment resulting from sensorimotor abnormalities that may affect all subsystems of speech including respiration, phonation, articulation, prosody, and resonance (*Duffy, 2013*). The dysarthrias are differentiated according to perceptual characteristics of speech and corroborated by the underlying neuropathology. In particular, PD is associated with hypokinetic dysarthria due to akinesia and bradykinetic-rigid syndromes, whereas HD shows hyperkinetic dysarthria resulting from chorea (*Duffy, 2013*). Despite the fact that both PD and HD are primarily disorders of the basal ganglia, the distinctive speech patterns connected with hypokinetic and hyperkinetic dysarthria are usually antagonistic. For instance, hypokinetic dysarthria in PD typically shows reduced vocal loudness and flattened loudness and pitch inflections, poor voice quality, variable and frequently increased speech rate, inappropriate silences and breathiness, while in contrast hyperkinetic dysarthria in HD demonstrates excess loudness and pitch variations, voice arrests, slow speech rate, inappropriate vocal noises and intermittent breathy segments (*Darley, Aronson & Brown, 1975*; *Logemann et al., 1978*; *Rusz et al., 2014*).

Interestingly, although hypokinetic and hyperkinetic dysarthria manifestations are often counteractive, hypernasality has been reported in both hypokinetic and hyperkinetic dysarthria (*Duffy, 2013*; *Hoodin & Gilbert, 1989*; *Chenery, Murdoch & Ingram, 1988*; *Logemann et al., 1978*; *Theodoros, Murdoch & Thompson, 1995*). In particular, investigation of both PD and HD provides us with the unique possibility to study the effect of basal ganglia dysfunction on the presence of hypernasality. Admittedly, hypernasality represents a distinctive manifestation of certain dysarthria subtypes, particularly of flaccid dysarthria, and thus its evaluation can provide useful information in the differential diagnosis of dysarthrias (*Duffy, 2013*).

Hypernasality is a result of velopharyngeal impairment and may be defined as the presence of inappropriate air leakage through the nasal cavity during phonation (*Warren, Dalston & Mayo, 1993*). This leakage may result from abnormal velopharyngeal structure, which is termed velopharyngeal insufficiency (VPI), and is present in patients with cleft palate, palatal fistula, and patients that have undergone maxillectomy. Other mechanisms

of hypernasality are distorted neuromuscular control of the levator veli palatini muscle and velopharyngeal seal, termed velopharyngeal incompetence (VIC), which includes patients with neurodegenerative diseases (*Folkins, 1988*). While abnormal velopharyngeal structure primarily leads to hypernasality, impaired neuromuscular control leading to dysarthria results in multiple speech distortions in which the particular effect of hypernasality may be less apparent to the listener due to the presence of other dysarthria manifestations. Thus, the majority of recent hypernasality research has been focused on VPI-induced hypernasality (*Dickson, 1962*; *Kataoka et al., 1996*; *Lee, Ciocca & Whitehill, 2003*; *Maier et al., 2008*; *Yoshida et al., 2000*), whereas only a few studies have investigated VIC hypernasality (*Hoodin & Gilbert, 1989*; *Chenery, Murdoch & Ingram, 1988*; *Poole et al., 2015*).

Studies examining hypernasality in PD have yielded controversial results. *Logemann et al. (1978)* perceptually detected hypernasality in only 10% of PD patients, whereas *Chenery, Murdoch & Ingram (1988)* and *Theodoros, Murdoch & Thompson (1995)* reported hypernasality in more than 30% of PD speakers. In addition, *Ludlow & Basich (1983)* included hypernasality among the 10 most distinctive perceptual features of PD, while *Darley, Aronson & Brown (1975)* did not find hypernasality to be a prominent feature of hypokinetic dysarthria. Considering HD speakers, to the best of our knowledge, no study has systematically examined hypernasality during hyperkinetic dysarthria, although *Duffy (2013)* reported intermittent hypernasality as one of the most deviant speech dimensions present in hyperkinetic dysarthria.

The etiology of hypernasality in PD and HD is unclear. Although the dysarthria is typically attributed to the disrupted motor control, little correspondence between speech and limb manifestations has been found (*Schulz & Grant, 2000*). Nevertheless, recent evidence based upon longitudinal follow-up data has shown that speech disorders in PD are generally related to the dopaminergic responsiveness of bradykinesia (*Rusz et al., 2016*). We may thus hypothesize that bradykinetic disturbances in soft palate control in PD may affect articulation of the velopharyngeal seal and accordingly lead to steady air leakage and increased hypernasality. Moreover, distorted neuromuscular control of levator veli palatini in PD may lead to increased hypernasality with increased fatigue during speech tasks.

In HD, the relationship between speech and limb manifestations appears to be more prominent. Correlation between speech timing parameters and overall motor disability has been noted previously (*Rusz et al., 2014*; *Skodda et al., 2014*). Furthermore, a relationship between laryngeal dysfunction and limb chorea has also been observed, likely as a result of laryngeal chorea (*Rusz et al., 2013*). Therefore, we hypothesize that choreatic movements of the velopharyngeal seal and velum may lead to varying resonance distortion, which would be in agreement with reported intermittent hyperkinetic dysarthria (*Duffy, 2013*).

Currently the most common method for hypernasality estimation is perceptual rating (*Kuehn & Moller, 2000*). In particular, perceptual assessment is considered the primary means to evaluate levels of nasality in children (*Vogel et al., 2009*). However, inter-rater and intra-rater reliability is questionable and perceptual rating requires a trained speech specialist (*Kuehn & Moller, 2000*). Consequently, more objective methods have been developed to complement perceptual ratings. Invasive methods, such as x-ray tracing with a lead pellet attached to the velum, provide direct observation of velopharyngeal movements

(*Hirose et al., 1981*). Other methods employ indirect estimation based on measurements of nasal airflow, nasal cavity sonography, nasometry comparing nasal and oral acoustic outputs, or the Horii Oral-Nasal Coupling Index (*Dillenschneider, Zaleski & Greiner, 1973*; *Hardin et al., 1992*; *Horii, 1980*). One of the least demanding methods with respect to patients and equipment is the 1/3-octave spectra, which is based on direct, non-invasive analysis of acoustic speech signal and was originally developed for the estimation of velopharyngeal insufficiency in cleft palate (*Kataoka et al., 1996*) and was later validated by *Vogel et al. (2009)*.

The 1/3-octave spectra method is a type of spectral analysis focused on the examination of spectral changes caused by resonatory speech pathologies. This method is based upon the linear source–filter theory of speech, which was first described by *Fant (1960)*. According to this theory, speech is partly created by a transfer function of the vocal tract. The introduction of the nasal cavity to the vocal tract leads to significant changes in its transfer function by incorporating nasal resonance Fn at an area around 1 kHz (*Stevens, 2000*). Several previous studies have shown that nasal resonance is a reliable marker of hypernasality (*Kataoka et al., 1996*; *Lee, Ciocca & Whitehill, 2003*; *Vogel et al., 2009*; *Yoshida et al., 2000*). However, some vowels may mask nasal resonance by the presence of formant frequencies in the area close to 1 kHz.

The vowel /i/ with the first formant frequency (F1) at approximately 240 Hz and the second formant frequency (F2) at approximately 2,400 Hz appear to be the most sensitive to nasal resonance (*Fant, 1960*; *Kataoka et al., 1996*; *Lee, Ciocca & Whitehill, 2003*; *Vogel et al., 2009*). Being the most evident, nasal resonance in the vowel /i/ should be more robust to anatomical variation of the nasal cavity including asymmetrical shape and varying shape of the connected sinuses. Moreover, the vowel /i/ is considered to be the most sensitive to nasal coupling (*Stevens, 2000*), and thus previous studies have focused on the quantitative evaluation of VPI hypernasality through the sustained vowel /i/ (*Kataoka et al., 1996*; *Lee, Ciocca & Whitehill, 2003*; *Yoshida et al., 2000*). Based on experiments with experienced listeners and the rating of nasality in artificially generated sounds in patients with cleft palate and those that underwent maxillectomy, previous studies have confirmed the vowel /i/ as an ideal speech task for hypernasality assessment (*Kataoka et al., 1996*; *Vogel et al., 2009*; *Yoshida et al., 2000*). Moreover, limited motion of the articulators including the jaw, tongue and lips in dysarthrias co-occur with velopharyngeal inadequacy and may play a more dominant role in changing the measures related to nasality. From this perspective, prolongation of vowel /i/ is a particularly suitable task to acoustically assess nasality in dysarthrias, as it represents relatively steady vocal function without the confounding effects of articulatory components of running speech.

Based upon these previous findings, the goal of the present study was to employ methods of objective hypernasality assessment and evaluate the presence and character of hypernasality in PD and HD speakers. A further aim was to examine possible relationships between the severity of hypernasality and disease-specific motor manifestations, to provide more insight into the pathophysiology responsible for development of hypernasality in basal ganglia disorders.

# METHODS

## Subjects

The participants in the present study were part of a larger investigation examining speech characteristics of patients with PD and HD. Previous reports generally focused on phonatory, articulatory and prosodic abnormalities including medication effects (*Rusz et al., 2013*; *Rusz et al., 2014*; *Rusz et al., 2016*). A total of 111 Czech native speakers, including 37 PD patients, 37 HD patients and 37 healthy participants were recorded.

The PD group consisted of 23 men and 14 women, mean age 63.1 ± 14.0 standard deviation (SD) (range 41–80) years, mean disease duration 8.0 ± 4.8 (1–24) years. All PD patients fulfilled the diagnostic criteria for PD (*Hughes et al., 1992*). All participants were on stable dopaminergic medication for at least 4 weeks before the examinations, which were conducted in the on-medication state. All PD patients underwent neurological examinations by an experienced neurologist and were rated according to the Hoehn & Yahr staging scale (H&Y, ranging from 1 to 5, where 1 indicates mild unilateral motor disorder and 5 indicates confinement to wheelchair or bed) and motor Unified Parkinson's Disease Rating Scale (UPDRS III, ranging from 0 to 108, with 0 for no motor manifestation and 108 representing severe motor distortion) (*Hoehn & Yahr, 1967*; *Stebbins & Goetz, 1998*). In addition, the UPDRS composite subscore of bradykinesia (sum of UPDRS III items 23, 24, 25 and 26, ranging from 0 to 24, with 0 for no bradykinesia and 24 representing severe bradykinetic distortion) was estimated (*Hughes et al., 1992*; *Jankovic, 2008*). Perceptual speech evaluation was based upon UPDRS III speech item 18 (range 0–4, with 0 representing normal speech and 4 indicating unintelligible speech). The H&Y score was 2.1 ± 0.4 (1–3), UPDRS III score was 17.5 ± 8.2 (4–36), the UPDRS bradykinesia subscore was 7.8 ± 3.6 (2–17), and the UPDRS III speech item 18 score was 0.8 ± 0.6 (0–2).

The HD group consisted of 19 men and 18 women with genetically confirmed HD with mean age 49.1 ± SD 12.7 (23–67) years, mean disease duration 6.1 ± 3.4 (1–16) years, mean number of CAG triplets 44.7 ± 3.3 (40–53). Most of the patients (32/37) were treated with monotherapy or a combination of benzodiazepines, antipsychotics, amantadine and antidepressants. All HD patients underwent extensive examination by an experienced neurologist and were rated according to the Unified Huntington's Disease Rating Scale (UHDRS, ranging from 0 to 124, where 0 indicates no motor disability and 124 indicates severe motor disability) (*Huntington-Study-Group, 1996*). In addition, the UHDRS chorea subscore was estimated (ranging from 0 to 28, where 0 indicates no motor disability and 28 indicates severe motor disability) (*Rusz et al., 2013*; *Walker, 2007*). Perceptual speech evaluation was based upon the UHDRS speech item (ranging from 0 to 4, where 0 indicates no disability and 4 indicates severe dysarthria). The UHDRS motor score was 25.7 ± 12.2 (3–54), the UHDRS chorea subscore was 8.6 ± 3.7 (0–14), and the UHDRS speech item was 0.8 ± 0.5 (0–2).

The healthy control (HC) group consisted of 23 men and 14 women, mean age of 63.1 ± 8.7 (41–77) years. None of the HC participants had a history of neurological or speech disorder. None of the HD, PD or HC subjects suffered from chronic obstructive pulmonary

disease, respiratory tract infection, allergy, asthma, facial paresis, or other malady that could negatively influence participant speech performance.

The study was approved by the Ethics Committee of the General University Hospital in Prague, Czech Republic, and all participants provided written, informed consent.

## Speech data

All recordings took place in a quiet room with a low ambient noise level using a head-mounted condenser microphone (Beyer-dynamic Opus 55, Heilbronn, Germany) positioned approximately 5 cm from each subject's mouth. The utterances were sampled at 48 kHz with 16 bit quantization. All the voice signals were obtained during single session conducted by a speech specialist, who asked participants to take a deep breath and perform sustained phonation of vowel /i/ at a comfortable loudness and pitch, as constant and long as possible. The measurement of sustained phonation was performed twice. The participants were also asked to provide freely spoken monologue on a given topic including family, work or interests, for at least two minutes. The both sustained phonation and monologue tasks were part of a comprehensive dysarthria test battery. No time limits were imposed during recording. The inclusion criteria were determined as the ability to sustain prolonged phonation for at least three seconds.

## Perceptual analysis

As connected speech is more demanding for velopharyngeal control, it is considered the most valid task for perceptual nasality estimation (*Kuehn & Moller, 2000*). The rating of nasality was based on speech material where the patient produced a monologue and performed by 10 raters including one speech-language pathologist, three clinicians and six acoustic speech specialists using a graded scale (0 = normal nasality, 1 = mild hypernasality, 2 = moderate hypernasality, 3 = severe hypernasality), based on The Great Ormond Street Speech Assessment '98 (GOS.SP.ASS.'98) (*Sell, Harding & Grunwell, 1999*). All the raters were trained by the speech-language pathologist prior to perceptual assessment. The perceptual assessment was performed blindly on randomized data consisting of all three participant groups. The presentation of samples was self-paced and performed by each rater separately, and each speech sample could be repeated at the discretion of the listener. The final score was obtained for overall perceptual rating across all raters by the median value computed from all perceptual assessments in the group. The inter-rater and intra-rater variability was estimated using a two way random average intra-class correlation (ICC). Intra-rater reliability was based upon the second perceptual assessments performed by all raters with more than three months delay. During the second assessment, each rater scored 27 randomly selected phonations (24% of entire dataset) equally representing PD, HD and HC groups.

## Acoustic analysis

For the purposes of instrumental analysis, two recording parts equal to 10% of signal length were cut off from both the beginning and end of the vowel /i/ to avoid distortion by initial vocal fold adjustment and fatigue at the end of the utterance. The remaining signal was then resampled to 20 kHz, which lowered the computational complexity and

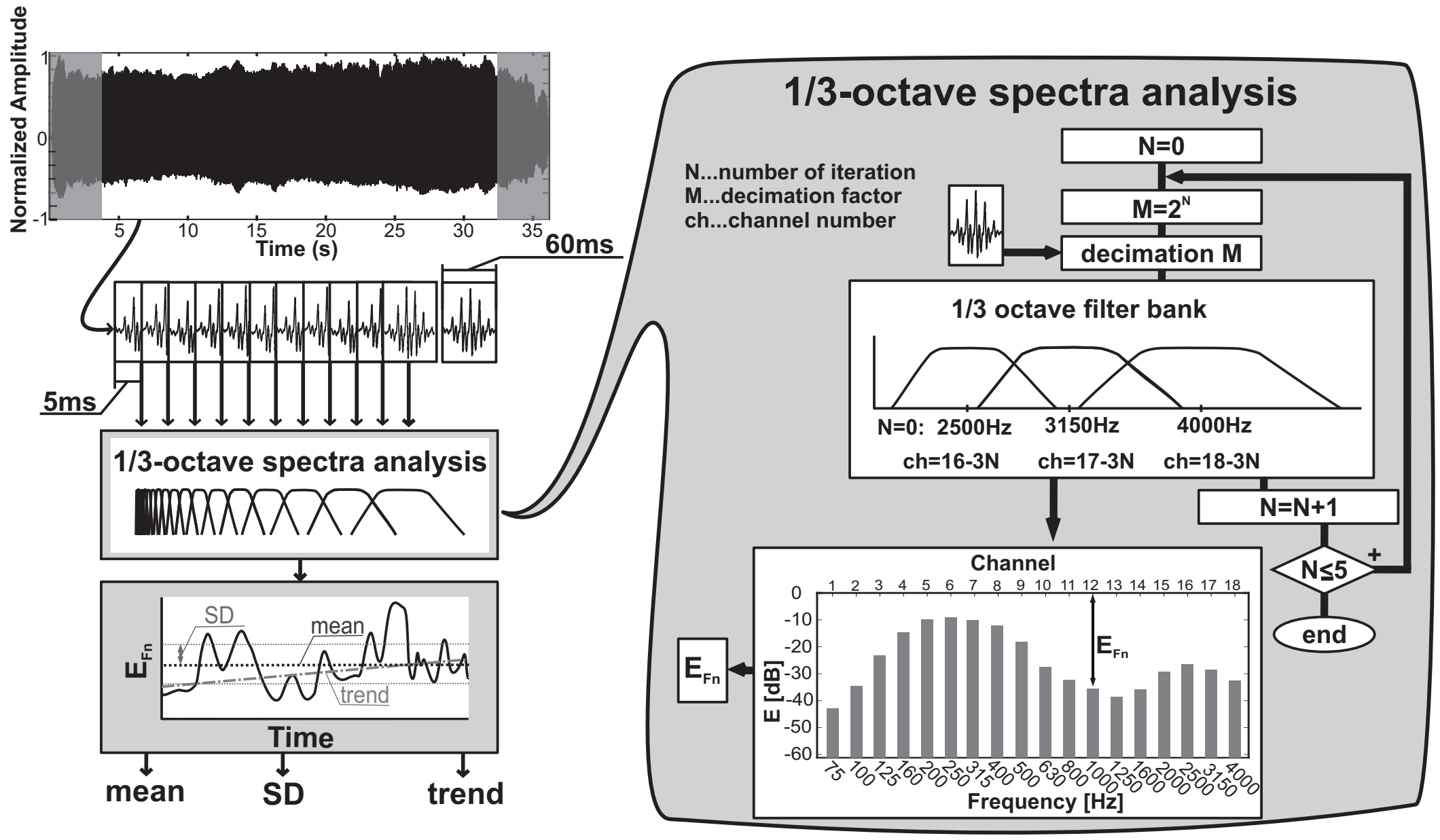

**Figure 1 Principle of acoustic analysis based on the 1/3-octave spectra assessment presented in *Couvreour (1998)*.**

preserved all useful information (*Titze, 1994*). The preprocessed signal was divided using a hamming 60 ms window with 55 ms overlap. Subsequently, each window was analyzed using a 1/3-octave spectra method.

The process of 1/3-octave spectra analysis based on the multirate filter bank presented by *Couvreour (1998)* is illustrated in Fig. 1. The three highest 1/3-octave frequency band filters were designed according to this method. For our purposes, the 3rd order IIR Butterworth filters were used and centered on octave frequencies of 2,500 Hz (passband from 2244.9 Hz to 2828.4 Hz), 3,150 Hz (passband from 2828.4 Hz to 3563.6 Hz), and 4,000 Hz (passband from 3563.6 Hz to 4489.8 Hz). After filtering, the highest components were removed from recording and the signal was then down-sampled by a factor of 2, i.e., sampling frequency (fs) to fs/2. Being defined in relation to the fs, the filter characteristics related to fs/2 yielded one octave lower for each down sampling. Based on this approach, the entire filter bank was achieved by the iterative use of signal down sampling. In each 1/3-octave frequency band, the root-mean-square (RMS) energy was estimated and achieved energy was transformed into decibels. A sum of energy contained in the entire 1/3-octave spectra was used as a reference value for the transformation into decibels, as described by Eq. (1).

$$E(i) = 10\log_{10}\left(\frac{E_{\text{filtered}}(i)}{\sum_{k=1}^{18} E_{\text{filtered}}(k)}\right), \tag{1}$$

where $E_{\text{filtered}}$ is energy contained in the single band of 1/3-octave and $E(i)$ is the decibel value of energy contained in the $i$th band.

Considering the effect of spectral flattening, nasality in sustained phonation of the vowel /i/ was evaluated using the $E_{\text{Fn}}$ parameter, which represented energy in a 1/3-octave band

centered around 1 kHz (passband from 890.9 Hz to 1122.5 Hz). This parameter reflected the addition of nasal resonance and additive nasal pole to the transfer function at 1 kHz. The overall level of hypernasality was estimated by the mean value of $E_{Fn}$ parameter ($E_{Fn}$ mean) across all windows in the entire utterance. The variability of nasality ($E_{Fn}$ SD) in speech was evaluated as the standard deviation of each parameter across the entire utterance. Finally, the evolution of hypernasality in the course of the utterance ($E_{Fn}$ trend) was described using a linear regression tangent for each parameter.

## Statistics

As the vowel /i/ was recorded twice for all speakers, average values of estimated acoustic parameters $E_{Fn}$ mean, $E_{Fn}$ SD and $E_{Fn}$ trend for each participant were used for all consecutive analyses.

The Kolmogorov–Smirnov test for independent samples was used to evaluate normality. Analysis of variance (ANOVA) with post-hoc Bonferroni adjustment was used for the estimation of group differences between PD, HD and HC groups across acoustic variables.

Relationships between variables were evaluated using Pearson's correlation and Spearman's correlation. Pearson's correlation was applied to normally distributed data (acoustic speech metrics and disease severity scores), whereas Spearman's correlation was used for non-normally distributed data (perceptual assessment of nasality and dysarthria severity). The Bonferroni adjustment for multiple comparisons was performed according to the four measures investigated ($E_{Fn}$ mean, $E_{Fn}$ SD, $E_{Fn}$ trend, and perceptual assessment) and the level of significance was set at $p < 0.0125$.

Due to the lack of information necessary for the classification of hypernasality, the assessment of the percentage of affected participants from acoustic data was based on the Wald task, which enables setting the classification specificity and sensitivity and therefore allows a more conservative threshold. The Wald task is a non-Bayesian statistical decision-making method which assumes that the dataset consists of two statistical distributions representing positive and negative cases and enables predefining false positive and false negative classifications by extending two basic classes (i.e., healthy and hypernasal), by an indecisive class (*Schlesinger & Hlavac, 2002*). Use of the indecisive class enables set boundaries where the possibility of a false positive or false negative result reaches a predefined value. Therefore, the indecisive class is used in cases where measured data do not provide sufficient information for clear-cut classification. In such cases, the user can decide whether the indecisive results would be discarded, incorporated with positive results providing the classifier with greater sensitivity and smaller selectivity or labeled as negative producing a less sensitive and more selective classifier. As a result, the method provides optimal cut-off values indicating if the subject already reached hypernasal speech performance or manifest normal nasality of wider norm of healthy speakers. In other words, the approach based on the Wald task avoids classifier overtraining and ensures certain confidence that cut-off values will be associated with hypernasal behavior. Comprehensive details on the Wald task have been published previously (*Rusz et al., 2011*).

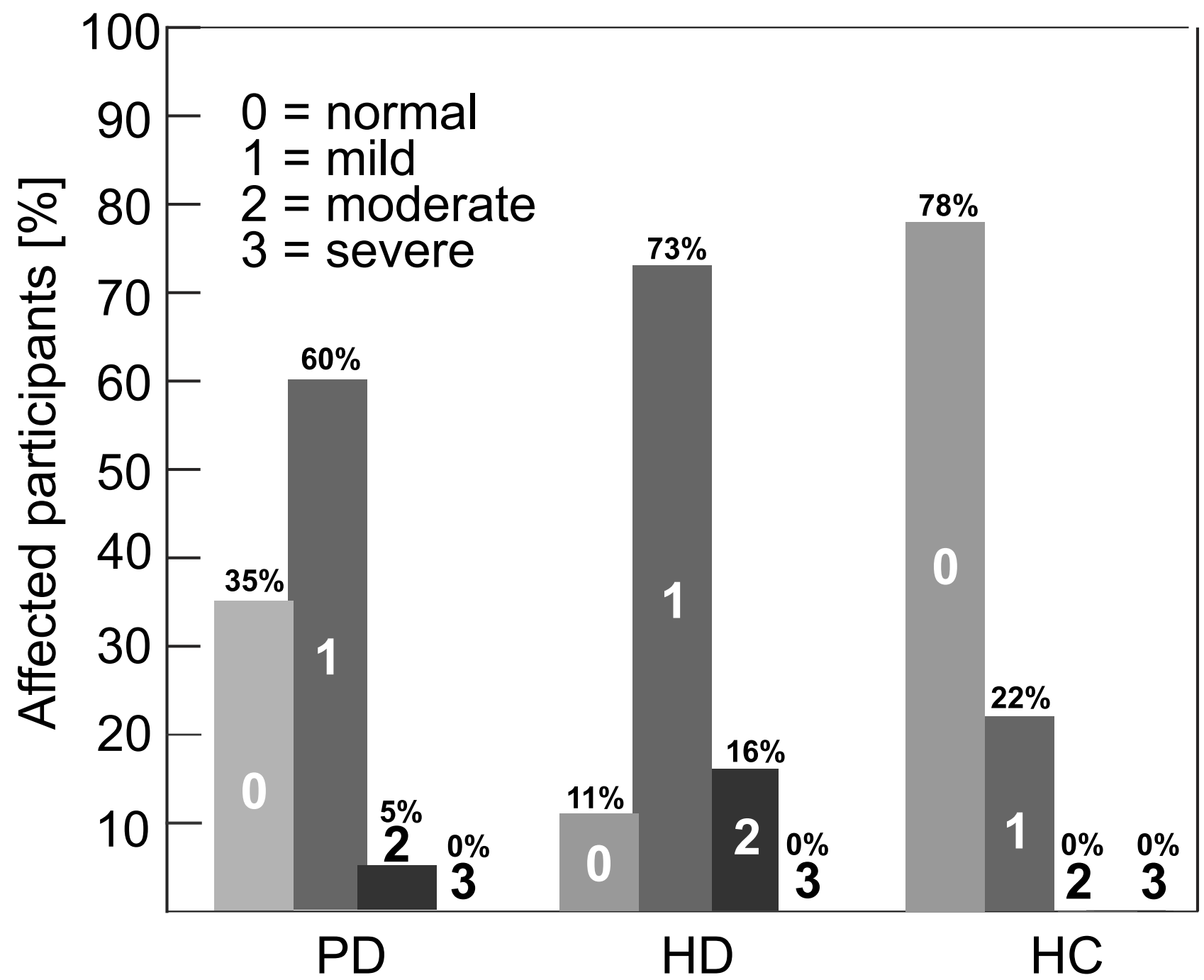

**Figure 2** **Percentage occurrence of hypernasality across participants according to the four grades perceptual score (0, no; 1, mild; 2, moderate; 3, severe) based on GOS.SP.ASS.'98 (*Sell, Harding & Grunwell, 1999*).**

# RESULTS

## Perceptual analysis

According to UPDRS III speech item 18, 10 PD patients (27%) demonstrated no speech impairment (score of 0), 32 PD patients (62%) mildly affected speech (score of 1) and 4 PD patients (11%) moderately affected speech (score of 2). According to the UHDRS speech item, eight HD patients (22%) showed normal speech (score of 0) and 29 HD patients (78%) dysarthria without the necessity of repeating speech to be intelligible (score of 1). In summary, the speech of all PD and HD patients was still fully understandable as indicated by UPDRS speech item 18 (ranging between 0 and 2) as well as the UHDRS speech item (ranging between 0 and 1).

The distribution of participants across four perceptual rating grades (no, mild, moderate, severe) are presented in Fig. 2. According to perceptual tests, 65% of PD and 89% of HD patients showed mild or moderate hypernasal speech performance, whereas mild hypernasality was observed in 22% of healthy speakers. The estimated inter-rater reliability was 0.85 ($p < 0.001$) across all raters and the intra-rater reliability ranged between 0.77 ($p < 0.05$) and 0.85 ($p < 0.001$) between individual raters.

## Acoustical analysis

Figure 3 illustrates the average energy distributions in PD, HD and HC groups across 18 frequency bands. As can be seen, the HD group demonstrates spectral flattening in the area between the F1 and F2 formant frequencies.

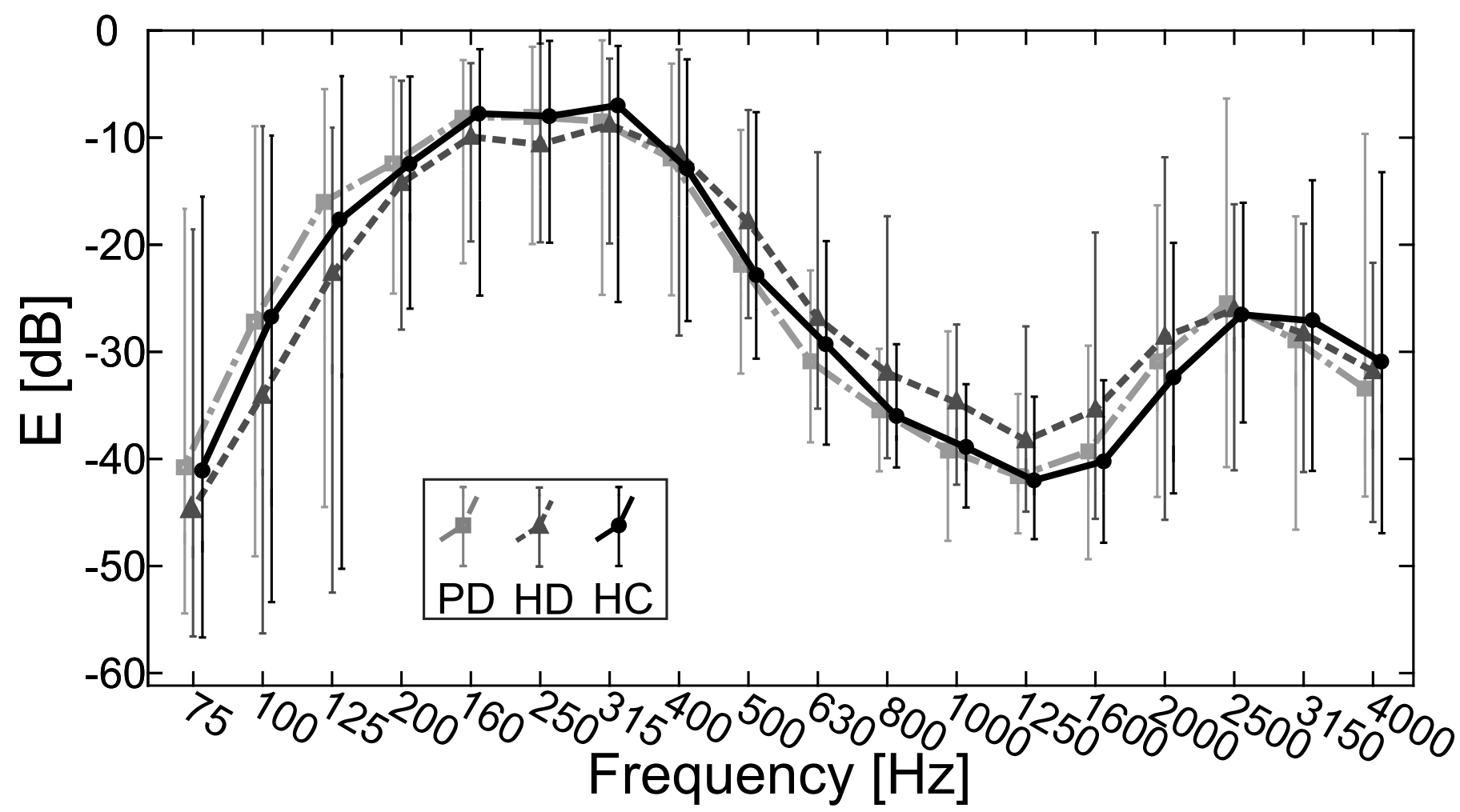

**Figure 3** **Measured average values of 1/3-octave spectra for 75–4,000 Hz bands with error bars indicating standard deviation for PD, HD and HC groups.**

**Table 1** **Results of hypernasality measures including mean and SD values for $E_{Fn}$ mean, $E_{Fn}$ SD and $E_{Fn}$ trend parameters across PD, HD and HC groups as well as results of ANOVA including $F$, $p$, and $\eta^2$ values.** Based upon post-hoc Bonferroni comparisons, an asterisk (∗) indicates statistically significant differences between HD and HC groups at the $p < 0.001$ level of significance.

| Measurement | PD | | HD | | HC | | ANOVA | | |
|---|---|---|---|---|---|---|---|---|---|
| | Mean | *SD* | Mean | *SD* | Mean | *SD* | $F(2,108)$ | $p$ | $\eta^2$ |
| $E_{Fn}$ mean (dB) | −38.93 | 4.37 | −34.85 | 4.59 | −39.10 | 3.06 | 11.82 | $p < 0.001$* | 0.179 |
| $E_{Fn}$ SD (dB) | 2.17 | 0.64 | 4.29 | 2.17 | 2.03 | 0.44 | 59.08 | $p < 0.001$* | 0.382 |
| $E_{Fn}$ trend (dB/s) | −4.784 | 18.58 | −2.22 | 82.32 | −3.68 | 17.76 | 0.21 | $p = 0.81$ | 0.000 |

Analysis of test–retest reliability of the proposed parameter $E_{Fn}$ showed strong correlation for mean ($r = 0.87$, $p < 0.001$) and SD ($r = 0.79$, $p < 0.001$) parameters, whereas trend analyses showed only moderate correlation ($r = 0.47$, $p < 0.001$). Table 1 lists the results of acoustic analyses. Statistically significant differences between all groups were observed for $E_{Fn}$ mean and $E_{Fn}$ SD ($p < 0.001$), particularly due to differences between HD and HC groups ($p < 0.001$).

Figures 4A–4C shows the percentage of affected participants according to Wald analysis. Using a cutoff value of −33 dB for $E_{Fn}$ mean, we found increased nasality in 27% of PD, 54% of HD and 19% of HC speakers. In addition, based upon a cutoff value of 3 dB for $E_{Fn}$ SD, we observed abnormal nasality variability in 27 % of PD, 78% of HD and 11 % of HC participants.

## Relationship between perceptual and acoustic analysis

Figure 4 shows comparisons related to the percentage of participants rated as hypernasal by acoustic methods and the overall perceptual score obtained across all raters for PD, HD, and HC groups. We observed significant correlation between overall perceptual rating and the acoustic $E_{Fn}$ SD parameter ($r = 0.51$, $p < 0.001$) but not $E_{Fn}$ mean parameter ($r = 0.09$, $p = 0.35$) or $E_{Fn}$ trend parameter ($r = 0.08$, $p = 0.38$).

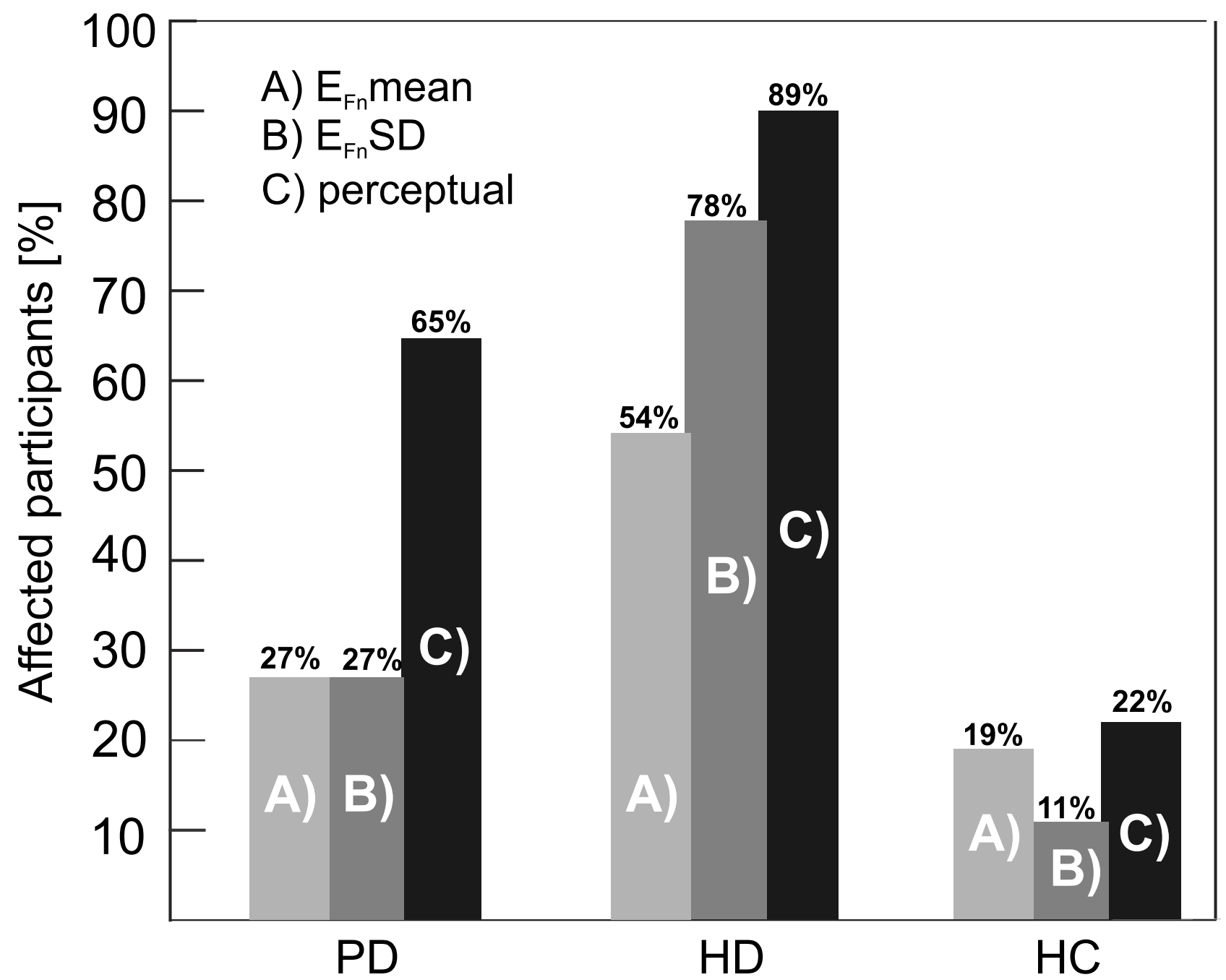

**Figure 4** Percentage of participants marked as hypernasal using (A) $E_{Fn}$ mean, (B) $E_{Fn}$ SD and (C) overall perceptual rating.

### Relationship between hypernasality and clinical manifestations

Table 2 lists results of correlations between hypernasality measurements and clinical manifestations for PD and HD groups. In the PD group, we did not detect any relationship between acoustic assessment of hypernasality and clinical metrics. In the HD group, we observed only significant relationships between the UHDRS chorea subscore and $E_{Fn}$ SD ($r = 0.42$, $p = 0.01$) and between UHDRS speech item and $E_{Fn}$ SD ($r = 0.46$, $p = 0.01$). We did not detect correlation between perceptual assessment and clinical manifestations in either PD or HD groups.

## DISCUSSION

In the present study, we analyzed hypernasality in PD, HD and HC utterances using objective acoustic analyses as well as perceptual assessment, which represents current gold standard for hypernasality evaluation. Based upon the 1/3-octave spectra analysis presented by *Kataoka et al. (1996)* and the acoustic model of the vocal tract published by *Fant (1960)*, we designed the parameter $E_{Fn}$ to evaluate the presence and character of hypernasality in prolonged vowels. Using acoustic analysis, we revealed an occurrence of hypernasality in 27% of PD, 54% of HD and 19% of HC speakers. In addition, our results showed a high occurrence of intermittent hypernasality in 78% of HD patients. Perceptual analysis showed the occurrence of mild to moderate hypernasality in 65% of PD, 89% HD and 22% HC speakers. Significant correlation between the acoustic parameter representing nasality fluctuation and perceptual assessment was observed. Furthermore, we revealed

**Table 2** Results of correlations between acoustical and perceptual measures of hypernasality and clinical manifestations of PD and HD groups.

| $r(p)$ | $E_{Fn}$ mean | $E_{Fn}$ SD | $E_{Fn}$ trend | Perceptual assessment |
|---|---|---|---|---|
| **PD** | | | | |
| UPDRS III | −0.10 (0.56) | 0.14 (0.41) | 0.04 (0.83) | −0.06 (0.74) |
| UPDRS III speech item 18 | −0.06 (0.75) | 0.26 (0.12) | 0.23 (0.18) | 0.27 (0.11) |
| UPDRS III bradykinesia subscore | −0.11 (0.50) | 0.15 (0.36) | 0.08 (0.62) | −0.05 (0.75) |
| Disease duration | 0.20 (0.24) | −0.32 (0.06) | −0.16 (0.34) | −0.06 (0.72) |
| **HD** | | | | |
| UHDRS | −0.01 (0.96) | 0.39 (0.02) | 0.23 (0.19) | 0.37 (0.03) |
| UHDRS speech item | −0.09 (0.59) | 0.46 (0.01) | −0.07 (0.70) | 0.16 (0.35) |
| UHDRS chorea subscore | 0.27 (0.12) | 0.42 (0.01) | 0.05 (0.76) | 0.08 (0.63) |
| Disease duration | 0.09 (0.60) | 0.28 (0.10) | 0.00 (0.99) | −0.05 (0.80) |

significant correlation between acoustic metric representing nasality fluctuation and chorea in HD patients.

## Nasality in PD

Although using acoustic analysis we detected hypernasality in 27% of PD speakers, the non-significant difference between PD and HC groups suggests that hypernasality is a non-prominent speech manifestation. Previous studies focused on hypernasality in PD have provided rather inconsistent conclusions. Based on perceptual evaluation, *Ludlow & Basich (1983)* included hypernasality among the 10 most salient features connected with dysarthria, whereas *Logemann et al. (1978)* observed hypernasality in only 10% of participants based on a large sample of PD patients. Considering instrumental analyses, only *Mueller (1971)* failed to detect hypernasality in PD speakers, contrary to the majority of studies reporting an increased occurrence of hypernasality in PD participants (*Hoodin & Gilbert, 1989*; *Netsell, Daniel & Celesia, 1975*; *Theodoros, Murdoch & Thompson, 1995*). While the differences in perceptual assessments could be explained by the fact that listeners from various cultures may have a different level of tolerance for perceived hypernasality, inconsistencies in the instrumental assessment are likely due to the differing sensitivity of particular methods. Moreover, both perceptual and instrumental assessment could be biased by differences in the sample data, as the majority of previous studies have reported hypernasality in a minority of PD speakers. One further explanation for these discrepancies may be that the severity of hypernasality parallels overall disease progression to some extent (*Hoodin & Gilbert, 1989*). However, we did not observe any relation between hypernasality metrics and disease duration, speech severity, or motor severity scales in PD.

## Nasality in HD

The presence of hypernasality was observed both perceptually and acoustically in the majority of our HD speakers, which was mainly associated with the occurrence of abnormal nasality variability. Indeed, we observed correlation between acoustic nasality variability and the chorea UHDRS subscore, demonstrating the significant impact of chorea on

velopharyngeal mechanism. Although our findings seem to be in accordance with *Duffy (2013)* that perceptually indicated intermittent hypernasality as a salient feature of patients manifesting chorea, there appear to be no other empirical data to support the results of the present study. Additionally, we also revealed relationship between acoustic nasality variability and overall dysarthria severity, indicating that the extent of abnormal nasality partially parallels increasing overall speech dysfunction in HD.

### Perceptual assessment of hypernasality

Previous studies have reported perceptual assessment of hypernasality in dysarthria as rather unreliable as hypernasality is less apparent to the listener due to the presence of more dominant dysarthria manifestations (*Brancewicz & Reich, 1989*). Nevertheless, although perceptual assessment of nasality in dysarthrias is challenging, it is still considered the gold standard, even in studies investigating acoustic techniques. Our results indicate more HD and PD participants systematically rated as hypernasal by perceptual assessment than by an instrumental approach, likely due to difficulty in achieving accurate perception of hypernasality when other abnormal dysarthria characteristics are present. Furthermore, the difference between speech tasks used during perceptual and instrumental evaluation could be a source of discrepancy between acoustic and perceptual assessments.

There is a little evidence for correlation between perceptual and instrumental measurements of hypernasality in dysarthrias (*Poole et al., 2015*; *Theodoros, Murdoch & Thompson, 1995*). In our HD sample, acoustic analyses identified only 50% of all HD speakers as hypernasal in comparison to the perceptual rating of nearly 90%. Yet, the abnormally intermittent character of nasality was also acoustically observed in nearly 80% of all HD participants. As we observed significant correlation between acoustic parameters measuring intermittent hypernasality and perceptual ranking, we may hypothesize that fluctuation in the level of nasality makes resonatory disruptions more obvious to perceptual raters. Interestingly, these correlations were evident even if perceptual and acoustic assessment were performed using different speech material.

In agreement with our findings, previous studies have perceptually rated the majority of PD participants as mildly hypernasal (*Hoodin & Gilbert, 1989*; *Theodoros, Murdoch & Thompson, 1995*). However, our raters tended to evaluate PD utterances with higher nasality scores in ambiguous cases. Indeed, some mild hypernasality is not rare even in healthy subjects and was observed in up to 22% of our control speakers, which is in accordance with previous research (*Poole et al., 2015*). Given this evidence, we may suppose that the perceptual decision between normal and mildly hypernasal speech can be misleading, particularly in dysarthrias with other perceptually dominant speech deviations.

### Acoustic assessment of hypernasality

In the present study, we applied an acoustic method designed for the objective evaluation of velopharyngeal insufficiency, to determine the presence and nature of velopharyngeal incompetence in PD and HD. This methodology has been previously found to be superior to other acoustic measures of hypernasality (*Vogel et al., 2009*), and was later successfully applied to patients with Friedreich ataxia resulting in velopharyngeal incompetence

(*Poole et al., 2015*). Based upon an acoustic model of the vowel /i/ published by *Stevens (2000)* and recommendations presented by *Kent et al. (1999)*, we designed the $E_{\mathrm{Fn}}$ parameter to describe the presence of nasal resonance in speech due to properties of the nasal cavity present in the 1 kHz 1/3-octave band (*Kataoka et al., 1996*; *Stevens, 2000*). This assumption is valid for all vowels; nevertheless, the wide plateau between F1 and F2 frequencies in the vowel /i/ makes the presence of nasal resonance more pronounced (*Kataoka et al., 1996*; *Stevens, 2000*). Compared to controls, the parameter $E_{\mathrm{Fn}}$ mean showed significantly increased energy in HD patients, suggesting an abnormal presence of hypernasality in HD patients. Furthermore, using the parameter $E_{\mathrm{Fn}}$ SD, we revealed significant differences in fluctuations of nasality between HD and control speakers, suggesting intermittent hypernasality in HD patients. The parameter $E_{\mathrm{Fn}}$ trend was found to be unreliable, as it demonstrated no significant differences between groups and low test–retest reliability.

### Limitations of the current study

We did not perform aerodynamic measurements, which would provide direct information about nasal airflow. Nevertheless, a previous study by *Vogel et al. (2009)* provided exhaustive evaluation of the 1/3-octave method and other studies have successfully applied this method to hypernasality assessment (*Kataoka et al., 1996*; *Lee, Ciocca & Whitehill, 2003*; *Poole et al., 2015*; *Yoshida et al., 2000*). The advantage of the current approach is that it provides an easy-to-administer acoustic assessment, which would be possible to integrate into a larger battery of acoustic tests.

It is noteworthy that the choice of the vowel /i/ may serve to maximize the impact of nasality or at least the likelihood of an acoustic model finding nasality. Thus, previous research on nasality in children used not only the optimal /i/ but a greater variety of speech material (*Vogel et al., 2009*). Therefore, the higher incidence of hypernasality, particularly in HD patients, due to a maximized impact of nasality cannot be excluded. Conversely, the results of perceptual tests suggest an even greater level of nasality across our participants than we were able to capture using acoustic assessment, indicating that level of nasality assessed using the 1/3-octave spectra method was not necessarily overestimated. Furthermore, the effect of maximizing nasality may be beneficial due to the fact that it emphasizes the presence of hypernasality among other dysarthria manifestations.

One limitation is that we used different speech tasks for the perceptual and acoustic evaluation of hypernasality, as accurate perceptual evaluation of hypernasality from sustained vowel phonation is not feasible. Indeed, the different speech tasks used likely make correlation analyses between perceptual and acoustic variables problematic. In future studies, it may therefore be beneficial to include rating for consistency, as with the Consensus Auditory Perceptual Evaluation of Voice (*Kempster et al., 2009*).

We did not test the consistency and reliability of UPDRS and UHDRS metrics. Nevertheless, relationships between nasality and motor abnormalities were found only for the UHDRS chorea subscore, which showed high inter-rater reliability with an ICC of 0.82 (*Huntington-Study-Group, 1996*).

As the presence of chorea in HD is unlikely to be limited only to specific parts of the vocal tract such as the soft palate, we cannot exclude that $E_{Fn}$ SD is also, to a certain extent, influenced by other manifestations of chorea, particularly laryngeal chorea (*Rusz et al., 2013*).

As HD generally has an earlier onset than PD, the PD and HD participant groups could not be age-matched. Therefore, we matched the age of the control group to the age of generally older PD group, as nasality is expected to remain stable throughout life or may slightly deteriorate as a consequence of aging (*Hoit et al., 1994*; *Ramig & Ringel, 1983*). This approach ensures that the results of the PD group were not favored in comparison with the HC group. Moreover, we did not match our groups according to gender. Nevertheless, previous studies did not find differences in nasality between male and female speakers (*Joos et al., 2006*; *Litzaw & Dalston, 1992*).

## CONCLUSION

Perceptual and acoustic data presented in the current study provide evidence of significantly increased and intermittent hypernasality in HD patients, presumably due to choreatic movements of the velopharyngeal mechanism. Although the presence of hypernasality was also observed in several PD speakers, abnormal nasality is not a prominent feature of hypokinetic dysarthria. However, further research is warranted. The relationships between proposed acoustic metrics and aerodynamic measurements for evaluation of hypernasality in dysarthrias should be explored. Future longitudinal studies are needed to confirm and further elaborate our findings and to show reliability of hypernasality measures as a possible marker of disease progression in basal ganglia disorders. Last but not least, as hypernasality is a prominent sign in several dysarthria subtypes (*Duffy, 2013*), sensitivity of methods proposed in the present study should be verified across various neurological disorders and measure of hypernasality may be useful in characterization of progressive neurological disorders as well as may have potential to provide important clues about the pathophysiology of underlying disease.

## ACKNOWLEDGEMENTS

Authors would like to thank to Vlastimila Čmejlová, Jan Hlavnička, Tomáš Lustyk, Veronika Stará, Vlastimila Stará, and Tereza Tykalová for perceptual ratings of hypernasality.

### Funding

This study was supported by the Ministry of Health of the Czech Republic, grant nr. 15-28038A. The funders had no role in study design, data collection and analysis, decision to publish, or preparation of the manuscript.

### Grant Disclosures

The following grant information was disclosed by the authors:
Ministry of Health of the Czech Republic: 15-28038A.

## Competing Interests

The authors declare there are no competing interests.

## Author Contributions

- Michal Novotný conceived and designed the experiments, performed the experiments, analyzed the data, contributed reagents/materials/analysis tools, wrote the paper, prepared figures and/or tables.
- Jan Rusz conceived and designed the experiments, performed the experiments, reviewed drafts of the paper.
- Roman Čmejla, Hana Růžičková, Jiří Klempíř and Evžen Růžička performed the experiments, reviewed drafts of the paper.

## Human Ethics

The following information was supplied relating to ethical approvals (i.e., approving body and any reference numbers):

The study was approved by the Ethics Committee of the General University Hospital in Prague, Czech Republic, and all participants provided written, informed consent.

## Data Availability

The raw data are supplied as Data S1 and the Matlab code for analysis is accessible through GitHub: https://github.com/MichalNovotny/hypernasality#hypernasality.

## Supplemental Information

Supplemental information for this article can be found online at http://dx.doi.org/10.7717/peerj.2530#supplemental-information.

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
