# Peer review of "Hypernasality associated with basal ganglia dysfunction: evidence from Parkinson’s disease and Huntington’s disease"

_PeerJ, doi:10.7717/peerj.2530_

## Round 0.1 · original submission · Minor Revisions

Dear authors

Apologies for the delay in providing reviews for your paper. As you will see, the reviewers are both very positive. Please address all the issues raised by Reviewer 1; Reviewer 2 refers to typos (and provides one example). Please correct it and also ensure that you have proof-read your manuscript before resubmission.

Reviewer 1 ·

Basic reporting

All seemsAppropriate

Experimental design

The description of participants is adequate. Data collection, perceptual analysis, and acoustic analysis have been implemented appropriately and are adequately described. The approach to assessment of nasality contributes to the literature on these disorders. Statistical tests are appropriate to the data.

Validity of the findings

I’d like to see data on correlations between disease severity, overall dysarthria severity and nasality scores, both acoustic and perceptual. This would strengthen the applicability of the methods and place them in context of the diseases as whole entities.

Could the authors explore the role of severity in the context of acoustic accuracy. Irrespective of diagnosis, how does the acoustic methods match up with moderate and severe ratings?

Additional comments

Another strong contribution from Rusz et al.
This article provides an acoustic and perceptual characterisation of nasal resonance in idiopathic Parkinson’s disease and Huntington’s disease. The article is clearly written, and adds novel analysis of speech disorder in these disorders, particularly Huntington’s disease. The article would be of interest to researchers of motor speech disorders.
Literature Review:
The relevant literature has been clearly discussed, and the purpose of the study have been clearly explained.
Methods:
- The Vogel et al. paper was verified in pediatric patients, with structural abnormalities – bot aspects different to the current groups. Perhaps this would affect sensitivity of the acoustic parameters.
- Further explanation of exactly which variables were examined for correlation would be useful (lines 288-291)
Results:
Results are clearly and accurately presented.
- Suggest that some mention of hypernasality be added to the caption of Figure 2
-
Discussion:
- The issues raised by the study have been adequately and clearly addressed. There are several limitations which have been well addressed by the authors, such as lack of correlation between acoustic and perceptual measures, and lack of findings of Hypernasality in the Parkinson’s disease group.
- What does “our raters tended to score PD speakers more strictly,” mean?
- Some additional discussion of whether the measures identified in the study should be further explored is warranted. The reader is left wondering whether there could be any clinical utility of the measures of nasalence for clinical diagnosis, or further characterisation of motor speech disorders in progressive neurological disorders.
- The Duffy textbook was referred to as though it included peer reviewed empirical data. Ideally the authors would find alternative primary sources to support findings.
Clarity:
- Line 85: suggest adding that Hypernasality is the presence of inappropriate air leakage through the nasal cavity during phonation. In order to distinguish from nasal aim emission.
- Line 107-109: suggest rewording the sentence “Although the dysarthria … manifestations has been found” , as the meaning is not immediately clear
- Line 177 & 194 : suggest introducing the fact that SD is given after the mean in the first instance on line 177, rather than line 194
- Line 341: Should Fig. 3 in fact be referring to Figure 4
- Line 438 – Friedreich ataxia is misspelled

Reviewer 2 ·

Basic reporting

Appropriate and sufficient information for readers.

Experimental design

Reasonable design and methodology for the study.

Validity of the findings

Data treated in an appropriate way and the findings reasonably stem from this.

Additional comments

typological errors e.g. 'hyfernasality'

---

## Round 0.2 · accepted · Accept

Thank you for submitting a revised manuscript. I am happy with the changes you have made in response to the comments of the reviewers.